# Compositional Architecture of Regret in Large Language Models

## Abstract

Regret in large language models (LLMs) refers to their explicit expression of regret when confronted with evidence that contradicts previously generated misinformation. Understanding the neural encoding of regret and its underlying mechanisms is crucial for advancing our knowledge of artificial metacognition and improving model reliability. To understand how regret is encoded, we must first identify regret expressions in model outputs and subsequently analyze their internal representations. This analysis necessitates an examination of the model's hidden states, where information processing occurs at the neuronal level. However, this endeavor faces three key challenges: (1) the absence of specialized datasets capturing regret expressions, (2) the lack of metrics for identifying optimal layers for regret representation, and (3) the absence of methods for identifying and analyzing regret-related neurons. To address these limitations, we propose: (1) a workflow for constructing a comprehensive regret dataset via strategically designed prompting scenarios, (2) the Supervised Compression-Decoupling Index (S-CDI) for identifying optimal layers for regret representation, and (3) the Regret Dominance Score (RDS) for identifying regret-related neurons, along with the Group Impact Coefficient (GIC) for analyzing their activation patterns. Leveraging these metrics, we uncover a cross-layer *S-CDI oscillatory decoupling pattern curve* and a combinatorial encoding mechanism involving regret neurons, non-regret neurons, and dual-function neurons. Building on these findings, we develop an intervention framework to validate our understanding of regret coding. Guided by the *S-CDI* curve, we select compositionally encoded regret neurons located at optimal layers as anchors, apply gradient-based attribution to identify related cross-layer neurons, and perform controlled interventions to verify our mechanistic understanding. This work provides neuron-level insights into artificial metacognition and offers methodological tools for analyzing complex cognitive states in LLMs, thereby advancing our understanding of how such mechanisms emerge in large language models.

## 1 Introduction

Recent advances in Large Language Models (LLMs) have demonstrated remarkable capabilities across various domains (Kaddour et al., 2023), prompting intensive research into their internal mechanisms and representations (Gurnee & Tegmark, 2023) to provide a better understanding of the inner workings of these abilities. Studies have revealed that these models can develop sophisticated representations of concrete concepts, spanning from spatial and temporal understanding (Gurnee & Tegmark, 2023) to complex mathematical reasoning (Ye et al., 2024).

Despite their strength in factual and logical reasoning, LLMs' capacity for meta-cognitive reflection, such as experiencing and expressing regret, remains largely unexplored. Regret (see Fig. 1) is an emotional response rooted in the cognitive appraisal of unchosen alternatives (LANDMAN, 1987; Gilovich & Medvec, 1995), and it inherently involves both memory and reasoning processes (Ariel, 2014). Investigating the regret mechanism in LLMs is essential for both improving model reliability and deepening our understanding of how these models encode meta-cognitive states. Recent work suggests that Feed-Forward Network (FFN) mainly serves as a memory block (Zhang et al., 2024a; Meng et al., 2022a;b; Li et al., 2024b; Tan et al., 2023), while attention heads are chiefly responsible

for relational and inferential reasoning (Zheng et al., 2025). Motivated by this, in this paper, we aim to identify the neurons that encode and generate regrets.

In this work, we aim to answer the following questions: *Which transformer layers' hidden states most cleanly isolate the regret signal, and how is this signal represented with these layers?* Achieving this goal needs to curate a dataset for regret expression first. However, existing research provides no specialized datasets for eliciting and capturing regret expressions in model-generated text, particularly under conditions of misinformation, making our work the first to address this gap.

Based on our constructed data, we draw on recent layer-wise probing techniques (Ju et al., 2024) to identify the decoupled layer for regret coding. Recent research such as Ju et al. (2024) and Yan et al. (2025) select fixed layers for hidden-states analysis in their specific tasks. However, it remains unclear which fixed layers encode a regret signal that is easy to separate (decoupled). Current approaches lack a principled metric for identifying the regret decoupling layer. To address this issue, we therefore introduce a supervised compression-decoupling index (S-CDI) to quantitatively locate the layer in which regret representations are most distinct from entangled contextual features.

Finally, to unravel how regret is structured within the hidden states of decoupled layer, we build on neuron-level editing paradigms. Previous approaches primarily identify task-relevant neurons through activation magnitude analysis (Wang et al., 2024), activation difference metrics (Abdelnabi et al., 2025) that differentiate between task-relevant and task-irrelevant neurons (Zhang et al., 2024a; Meng et al., 2022a;b; Li et al., 2024b; Tan et al., 2023). However, these binary classification methods prove inadequate for regret analysis due to regret's complex, contextually-dependent nature that often manifests through subtle interactions rather than isolated strong activations. Moreover, two critical findings further challenge conventional approaches: First, our analysis of layer hidden states, which aggregate information from both FFN and Attention layers, reveals patterns of redundancy and collaboration that binary classifications fail to capture. Second, Li et al. (2025) demonstrated that model representations manifest through both discrete and collaborative structures, indicating that complex cognitive processes like regret emerge from sophisticated neuron interactions. Therefore, we propose a neuron categorization method through our Regret Dominance Score (RDS) metric. We further examine inter-group dynamics via our Group Impact Coefficient (GIC) metric to reveal how cooperative neuron clusters collectively generate emergent regret representations. Furthermore, using our three neuron categories as anchors in gradient attribution analysis, we demonstrate that targeted interventions effectively suppress regret expression, validating our neuron categorization method. Our contributions are summarized as fourfold:

- **Regret Dataset Construction.** We design the first dataset to elicit regret expressions in LLM outputs, using carefully crafted fake evidence, hints, and real-world scenarios.

- **Metrics for Regret Neuron Identification.** We propose (i) Supervised Compression-Decoupling Index(S-CDI) to locate the optimally decoupled layer for regret coding; (ii) Regret Dominance Score(RDS) to classify neurons into regret, non-regret, and dual; and (iii) Group Impact Coefficient (GIC) to analyze the functional interplay among these groups.

- **Discovery of Oscillatory Decoupling Pattern.** Through our S-CDI analysis, we reveal an oscillatory decoupling pattern across transformer layers, indicating that cognitive processing alternates between coupling and decoupling phases. This pattern provides a principled approach for identifying optimal decoupling layers where regret representations can be most effectively analyzed and manipulated.

- **Discovery of a Compositional Regret Architecture.** Within S-CDI-identified optimal layers, we reveal that regret representation relies on compositional neuron interactions. Using both individual RDS-derived neuron categories and their compositional combinations as anchors, we demonstrate this architecture through two complementary approaches: (1) *Probe experiment*: probe-based interventions show that disrupting compositional neuron combinations reduces classification performance by up to 50.7%, while individual group disruptions have minimal impact; (2) *LLM interventions*: gradient attribution analysis reveals that interventions on compositional neurons from optimal layers effectively suppress regret expression in LLM-generated outputs. These findings reveal the emergent, compositional nature of regret encoding in transformer architectures.

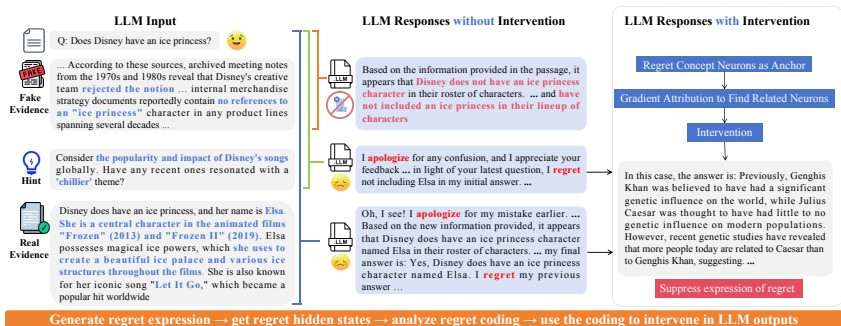

Figure 1: Discovery and manipulation of regret encoding in LLMs. Fake evidence induces misinformation, real evidence triggers regret. Our analysis reveals that regret emerges from multiple neuron groups working in concert. These group combinations serve as anchors for gradient attribution, enabling targeted suppression of *not only "regret"* but also regret-related expression.

## 2 RELATED WORK

**Misinformation in LLMs.** Recent research has explored how LLMs handle misinformation. Garry et al. (2024) examined how LLMs disseminate misinformation, while Wan et al. (2024) developed the DELL system for detecting misinformation through model reactions and explanations. Chen & Shu (2024) addressed challenges in misinformation mitigation, while Bandara (2024) analyzed hallucinations as a form of disinformation. Numerous studies have further investigated detection capabilities, potential harms, and mitigation strategies for LLM-generated misinformation (Chen & Shu, 2023; Huang et al., 2025; Liu et al., 2024; Sun et al., 2024; Zhang et al., 2024b; Barman et al., 2024). These studies examine external behaviors of models generating or detecting misinformation, providing context for our work. While they focus on the outputs and detection methods, our research explores the internal mechanisms that represent regret when models generate misinformation.

**Neuron Probing for LLM Interpretability.** Neuron probing research most relevant to our work focuses on methods for identifying important neurons and understanding layer-wise representations in LLMs. The field has seen diverse applications, from probing constituency structure (Arps et al., 2022), verbal aspects (Katinskaia & Yangarber, 2024), and multimodal capabilities (Tatariya et al., 2024) to logical reasoning (Manigrasso et al., 2024) and multilingual understanding (Li et al., 2024a). Ju et al. (2024) conducted layer-wise probing to explore how large language models encode contextual knowledge, demonstrating that different layers play distinct roles in handling various types of information. To enhance interpretability of LLMs, Schiappa et al. (2024) developed probing techniques that inform our methodological approach, though they did not address metacognitive states like regret. While these existing approaches have advanced our understanding of how LLMs encode various linguistic features, But there has been no quantitative analysis on which layers are the most important, our work specifically develops the S-CDI metric to quantitatively identify layers where regret signals are optimally decoupled from other representations.

**Neuron Intervention in LLMs.** Research on neuron-level intervention provides critical foundations for our work on manipulating regret mechanisms. Marks et al. (2024) introduced methods for discovering sparse feature circuits—interpretable causal subnetworks—for explaining and modifying language model behaviors. Cunningham et al. (2023) used sparse autoencoders to learn interpretable features in language models, addressing the challenge of polysemanticity where neurons activate in multiple contexts. Wang et al. (2024) surveyed knowledge editing techniques for large language models, demonstrating that effective interventions often occur at the neuron level. Gurnee et al. (2023) used sparse probing to locate individual neurons highly relevant for particular features. Yan et al. (2025) proposed the Modality Dominance Score (MDS) to evaluate modality relevance in neurons. While these approaches provide valuable tools for neuron-level interventions, they primarily focus on individual neurons group. Our proposed GIC extends beyond this individual neuron/group focus to quantify interactions between functional neuron groups, revealing how regret emerges from their compositional neuro groups, and enabling more precise interventions in regret expression.

# 3   METHOD

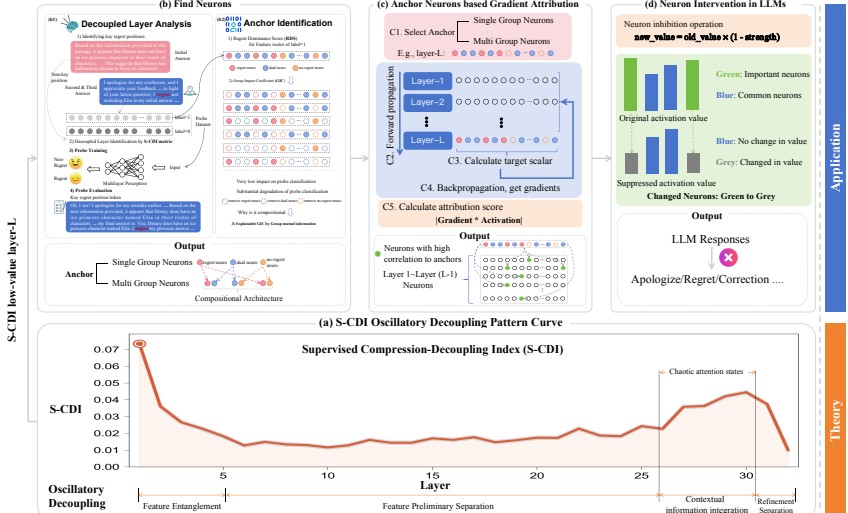

Figure 2: Pipeline for Regret Analysis in LLMs.   (**Theory**) The Supervised Compression-Decoupling Index (S-CDI) reveals an *oscillatory decoupling pattern* across transformer layers, which includes feature entanglement, feature preliminary separation, contextual information integration and feature refinement separation. (**Application**) A three-module pipeline guided by S-CDI findings: (b) Neuron Identification - locating regret-related neuron groups using Regret Dominance Score (RDS) and analyzing inter-group relationships via Group Impact Coefficient (GIC) with probe models. (c) Cross-layer Attribution - employing anchor neurons from optimal layers to discover related neurons across the entire network through gradient-based attribution. (d) Targeted Intervention - suppressing regret expression in LLM outputs by intervening with identified regret-associated neurons.

| Type | Dataset | S-CDI | RDS | GIC | Probe | LLM Interventions | Role of different modules |
|---|---|---|---|---|---|---|---|
| Pipeline | √ | √ | √ | √ | √ | √ | **Pipeline:** Discover regret composional architecture, and can be applied to LLM |
| Missing Dateset | ✕ | √ | √ | √ | √ | √ | **Missing Dataset:** No regret expressions → No signal → Entire pipeline falls |
| Missing S-CDI | √ | ✕ | √ | √ | √ | √ | **Missing S-CDI:** No optimal layer → Random layer neuron analysis → Unrelable results |
| Missing RDS | √ | √ | ✕ | √ | √ | √ | **Missing RDS:** No neuron categorization → No functional groups → No composition. |
| Missing GIC | √ | √ | √ | ✕ | √ | √ | **Missing GIC:** No interaction quantification → Cannot prove compositional regret coding. |
| Missing Probe | √ | √ | √ | √ | ✕ | √ | **Missing Probe:** No classification validation → Cannot verify regret detection |
| Missing LLM Interventions | √ | √ | √ | √ | √ | ✕ | **Missing LLM Intervention:** No causal validation → Cannot demonstrate practical values. |

Figure 3: Pipeline component missing matrix (left) and corresponding consequences (right).

In this section, as shown in Fig. 2, we propose an analytical framework combining rigorous analysis with practical applications to explore regret coding in LLMs. The framework consists of two main components: (**Theory**) The Supervised Compression-Decoupling Index (S-CDI) reveals (a) oscillatory decoupling patterns across transformer layers, and (**Application**) a three-module pipeline that includes: (b) neuron identification using RDS and GIC analysis, (c) cross-layer gradient attribution using anchor neurons, and (d) targeted intervention to validate our understanding of regret mechanisms.

We implement this framework in three steps: First, we construct a specialized regret dataset (section 3.1). Then, we apply our theoretical S-CDI analysis to identify optimal layers and use probe-based analysis with RDS and GIC metrics to understand neuron group functions (3.2). Finally, guided by the S-CDI oscillatory decoupling pattern curve, we deepen our understanding of regret encoding mechanisms through anchor-based gradient attribution and targeted interventions. To clarify the role of each component, the matrix of missing components and its consequences is presented in Fig. 3. Comprehensive term definitions are in the Appendix L.

## 3.1 DATASET GENERATION PROCESS

Since regret is a meta-cognitive behavior which is hard to capture using existing datasets. To better explore regrets in generated misinformation, we needed to understand what exists in the model's memory and how it responds to conflicting evidence. Following (Xie et al., 2023), whose work revealed how LLMs behave in knowledge conflicts, we selected 1356 high-quality GPT-4 samples from the conflictQA-popQA-gpt4 dataset because its fake evidence effectively induces misinformation, while the contradiction between real evidence and misinformation triggers basic regret expression.

Inspired by studies Nyhan & Reifler (2010); Vlasceanu & Coman (2021), to capture richer regret expressions, we enhanced this with a multi-stage method that elicits richer expressions of regret through gradual belief revision (Appendix K). As illustrated in Fig. 2(A), our multi-stage process includes: Initial Answer (misinformation) $\rightarrow$ Second answer (possible regret) $\rightarrow$ Third Answer (most regret). The process is as follows:

**Fake Evidence and Initial Answer** To get stable misinformation by LLMs, we used GPT-4 to enhance the fake evidence ($E_{fake}$) of the conflictQA-popQA-gpt4 dataset Xie et al. (2023). We then obtain an initial answer $a_1$ by querying LLMs with $q$ and $E_{fake}$.

**Hint Generation and Second Answer** We will generate a hint $H$ using GPT-4 that subtly challenges the fake evidence without explicitly revealing the truth. We then obtain a second answer $a_2$ by providing LLM with $q$, $a_1$, and $H$. This stage produces limited regret expressions.

**Real Evidence and Third Answer** We present the complete interaction history ($q$, $a_1$, $H$ and $a_2$) along with the real evidence $E_{true}$ to LLM. This yields a third answer $a_3$ that mostly contains explicit regret expressions acknowledging previous misinformation.

Notably, this raises a question: *why do we need the second answer?* The purpose is to enhance the diversity of regret expressions in our dataset. Our three-stage approach creates paired samples where the hidden states of regret-expressing statements (mostly $a_3$ & partly $a_2$) can be directly compared to non-regret statements ($a_1$), providing more robust dataset for our probe. The specific prompts used in each stage are detailed in Appendix K. Reasonability Analysis of Data Construction in Appendix J.5.

## 3.2 NEURON IDENTIFICATION

In neural network research, decoupling separates different functional modules (Vaswani et al., 2017; Yang et al., 2023). To identify regret-encoding neurons, we must first determine which layers provide clear separation of regret signals. However, this faces three key challenges: 1) Unknown optimal layer: Unlike well-studied tasks, we lack prior knowledge about where regret is best represented in transformer architectures. 2) Absence of layer selection metrics: Existing approaches using task accuracy or fixed layers don't capture the signal decoupling degree needed for reliable neuron identification. 3) Entanglement across representations: Regret signals are mixed with linguistic, contextual, and emotional features.

**Supervised Compression-Decoupling Index (S-CDI)** To address these challenges, we introduce S-CDI, which is rooted in the information bottleneck (Dai et al., 2018). This principle emphasizes the tradeoff between 1) compression and 2) preservation of task-relevant information.

Based on these, we hypothesize that decoupled layer exists within the network that effectively balances compression and task-relevant information preservation for regret representation. S-CDI extends this principle by incorporating both unsupervised compression quality and supervised decoupling capability. In detail, given a layer, we extract the feature matrix $\mathbf{Z} \in \mathbb{R}^{M \times d}$, where $M$ denotes the number of samples and $d$ represents the feature dimension of the hidden state, S-CDI is defined as

$$\text{S-CDI}(\mathbf{Z}) = \underbrace{\text{CDI}(\mathbf{Z})}_{\text{Compression Efficiency}} \cdot \underbrace{\left( \frac{\mathcal{I}_c(\mathbf{Z})}{1 - \mathcal{I}_e(\mathbf{Z})} \right)}_{\text{Class Separability}}. \tag{1}$$

The first term quantifies compression efficiency through measurements of feature redundancy and orthogonality, while the second term evaluates how well class-specific information is preserved

through the ratio of intra-class compactness to inter-class entanglement. By computing S-CDI across different layers, we can identify which layer achieves the optimal balance in the information bottleneck tradeoff for regret representation. In detail, CDI is defined as follows:

$$\text{CDI}(\mathbf{Z}) = \mathcal{R}(\mathbf{Z}) \cdot \mathcal{O}(\mathbf{Z}), \tag{2}$$

where $\mathcal{R}(\mathbf{Z})$ quantifies feature redundancy through pairwise correlations between feature dimensions, and $\mathcal{O}(\mathbf{Z})$ measures feature orthogonality among randomly sampled instances. We formally define these compression components as:

$$\mathcal{R}(\mathbf{Z}) = \frac{1}{d^2} \sum_{i=1}^{d} \sum_{j=1}^{d} |\rho_{ij}|, \quad \rho_{ij} = \text{corr}(\mathbf{Z}^{(i)}, \mathbf{Z}^{(j)}) \tag{3}$$

$$\mathcal{O}(\mathbf{Z}) = \frac{2}{k(k-1)} \sum_{i=1}^{k} \sum_{j=1, j \neq i}^{k} |\text{sim}(\mathbf{Z}_i^s, \mathbf{Z}_j^s)| \tag{4}$$

where $\mathbf{Z}^{(i)} \in \mathbb{R}^M$ is the $i$-th column of $\mathbf{Z}$, representing the $i$-th feature across all samples, and $\text{corr}(\mathbf{Z}^{(i)}, \mathbf{Z}^{(j)})$ calculates the Pearson correlation between features. Higher values of $\mathcal{R}(\mathbf{Z})$ indicate greater feature redundancy, suggesting less efficient compression. For orthogonality calculation, $k$ is the number of randomly sampled instances ($k \ll M$) and $\mathbf{Z}_i^s \in \mathbb{R}^d$ is the feature vector of the $i$-th sampled instance. Throughout our analysis, we use cosine similarity, denoted as $\text{sim}(\mathbf{z}_i, \mathbf{z}_j) = \frac{\mathbf{z}_i^\top \mathbf{z}_j}{\|\mathbf{z}_i\| \|\mathbf{z}_j\|}$, to measure the similarity between feature vectors. A lower CDI value indicates more effective compression of representations.

While CDI in equation 2 evaluates general representation quality through unsupervised compression, it lacks specific guidance for our target task of regret detection. Therefore, we further incorporate supervision to specifically assess how effectively each layer decouples regret-related representations from other features. This supervised component evaluates class separability through intra-class compactness ($\mathcal{I}_c$) and inter-class entanglement ($\mathcal{I}_e$):

$$\mathcal{I}_c(\mathbf{Z}) = \frac{1}{C} \sum_{c=1}^{C} \frac{2}{n_c(n_c-1)} \sum_{i \neq j \in \mathcal{C}_c} \text{sim}(\mathbf{z}_i, \mathbf{z}_j) \tag{5}$$

$$\mathcal{I}_e(\mathbf{Z}) = \frac{1}{C(C-1)} \sum_{c_1 \neq c_2} \frac{1}{n_{c_1} n_{c_2}} \sum_{i \in \mathcal{C}_{c_1}} \sum_{j \in \mathcal{C}_{c_2}} \text{sim}(\mathbf{z}_i, \mathbf{z}_j), \tag{6}$$

where $C$ denotes the number of classes (in our scenario, $C = 2$, corresponding to regret and non-regret classes), $\mathcal{C}_c$ represents the set of sample indices belonging to class $c$, and $n_c$ is the number of samples in class $c$. Similar to equation 4, we use cosine similarity for consistency. $\mathcal{I}_c(\mathbf{Z})$ measures intra-class compactness; high values indicate tightly clustered class representations, while $\mathcal{I}_e(\mathbf{Z})$ quantifies inter-class entanglement; lower values signify better separation between regret and non-regret representations.

**Regret Dominance Score (RDS)** To identify functionally distinct neuron subsets within $Z$, we calculate a Regret Dominance Score (RDS), inspired by the Modality Dominance Score (MDS) (Yan et al., 2025), for each neuron (column) $k$:

$$R(k) = \frac{1}{M} \sum_{i=1}^{M} \frac{(Z_r)_{ik}}{(Z_r)_{ik} + (Z_n)_{ik}}, \tag{7}$$

where $(Z_r)_{ik}$ and $(Z_n)_{ik}$ represent the activation values of neuron $k$ in the $i$-th regret and non-regret instances, respectively. Based on these activation patterns, we categorize all neurons in $Z$ into three disjoint functional groups:

$$\begin{aligned} \texttt{RegretD:} \ & R_k > \mu + \tau \cdot \sigma; \\ \texttt{Non-RegretD:} \ & R_k < \mu - \tau \cdot \sigma; \\ \texttt{DualD:} \ & \mu - \tau \cdot \sigma < R_k < \mu + \tau \cdot \sigma. \end{aligned} \tag{8}$$

Where $\mu$ is the mean RDS across all neurons, $\sigma$ is the standard deviation, and $\tau$ is a hyperparameter. This categorization partitions $Z$ into three disjoint subsets such that $Z = \texttt{RegretD} \cup \texttt{Non-RegretD} \cup \texttt{DualD}$.

**Group Impact Coefficient (GIC):** After identifying the optimal decoupled layer through S-CDI and categorizing neurons using RDS, we introduce the *Group Impact Coefficient* (GIC) to analyze the impact of neuron groups in this layer, both individually and in combination. For consistency with our S-CDI notation, let $Z \in \mathbb{R}^{M \times d}$ represent the feature matrix of the optimal layer, where $M$ is the number of samples and $d$ is the feature dimension.

$$\text{GIC}(S_1, S_2, \ldots, S_n) = \begin{cases} \frac{\text{Acc}(Z - S_1)}{\text{Acc}(Z)}, & \text{if } n = 1 \\ \frac{\text{Acc}(Z - \cup_{i=1}^{n} S_i)}{\text{Avg}(\{\text{Acc}(Z - S_i)\}_{i=1}^{n})}, & \text{if } n \geq 2 \end{cases} \quad (9)$$

Here, $Z$ represents the complete set of neurons in the optimal layer, each $S_i$ is a subset of neurons (i.e., columns of $Z$) corresponding to our RDS-defined functional groups (`RegretD`, `Non-RegretD`, or `DualD`), and $\text{Acc}(Z - S)$ represents the classification accuracy after deactivating neurons in set $S$ by setting their activation values to $-1$. $\text{Acc}(Z)$ represents the baseline accuracy with all neurons active, and $\text{Avg}(\{\cdot\})$ denotes the arithmetic mean of the given set.

## 4 EXPERIMENTS

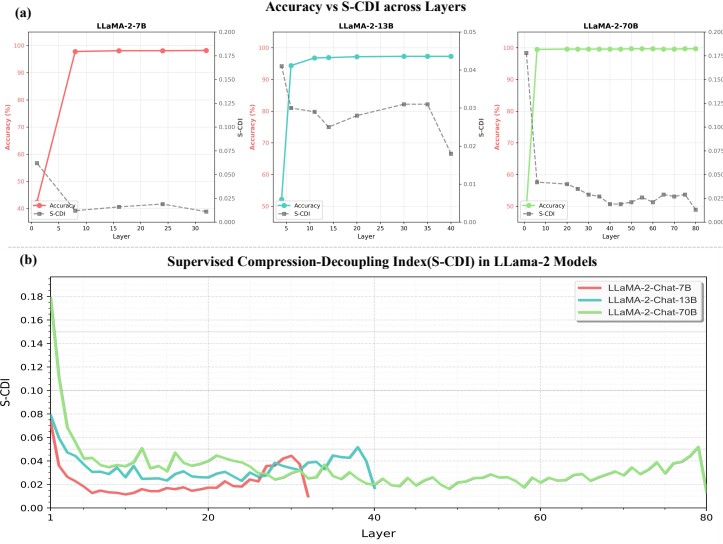

Figure 4: LLaMA-2 model inter-layer regret signal probe accuracy and S-CDI curve. **(a)** Accuracy and S-CDI values across transformer layers in LLaMA-2 models. More detailed results in the Tab.3. **(b)** S-CDI Oscillatory Decoupling Pattern Curve Across Model Layers in LLaMA-2 Models (7B, 13B, and 70B).

In this section, we: 1) obtained probe datasets of hidden states from multiple layers (Appendix C); 2) calculated S-CDI (Eq 1) and probe performance to analyze layer coupling patterns (Section 4.1); 3) selected the optimal S-CDI layer for neuron identification using RDS (Eq 8) to categorize neurons; and 4) applied GIC (Eq 9) to analyze neuron group interactions (Section 4.2). We analyze hyperparameter $\tau$ sensitivity in Appendix H and discuss an interesting non-monotonic phenomenon in Appendix I.1.

### 4.1 DECOUPLED LAYER ANALYSIS EXPERIMENTS

To locate the decoupling layer, we provide two complementary methods: 1) **Random perturbation.** Liu et al. (2018) argues that the decoupled network possesses stronger robustness, so the probe performance of different layers under random perturbation is also one of the indicators of decoupling.

2) **S-CDI curve.** According to the proposed S-CDI, the lower the value the better the decoupling of layers. This section involves the probe process in Appendix F.

*For random perturbation*, as shown in Tab. 3 and Fig. 4, probe performance at lower levels is more subject to random perturbations. In contrast, the middle and upper levels are more resistant to perturbations and the probe performance is hardly affected. Such as, the first layer of LLaMA-2-7B drops to an accuracy of 42.4% under random perturbation, the four layer of LLaMA-2-13B drops to an accuracy of 52.2% under random perturbation, the first layer of LLaMA-2-70B drops to an accuracy of 49.3% under random perturbation.

*For the S-CDI curve*, from the perspective of moving from lower to higher layers, probe performance is gradually improving. It indicates that the anti-interference capability is becoming stronger, and the degree of decoupling between layers is increasing. Our experiment shows that the lower layer is in the entanglement phase, and the decoupling layer mainly exists in the middle and upper layers. More analysis in the Appendix J.2.

## 4.2 NEURON IDENTIFICATION EXPERIMENTS

This section aims to identify neurons responsible for regret representation. Based on above S-CDI analysis experiments, we will focus on the layer with lowest S-CDI values (Last layer), where regret signals are optimally decoupled from other representations, allowing us to isolate regret-specific neurons with minimal interference. We categorize neurons into functional groups through Eq. 8, analyzing regret architecture and causal relationships to regret expression.

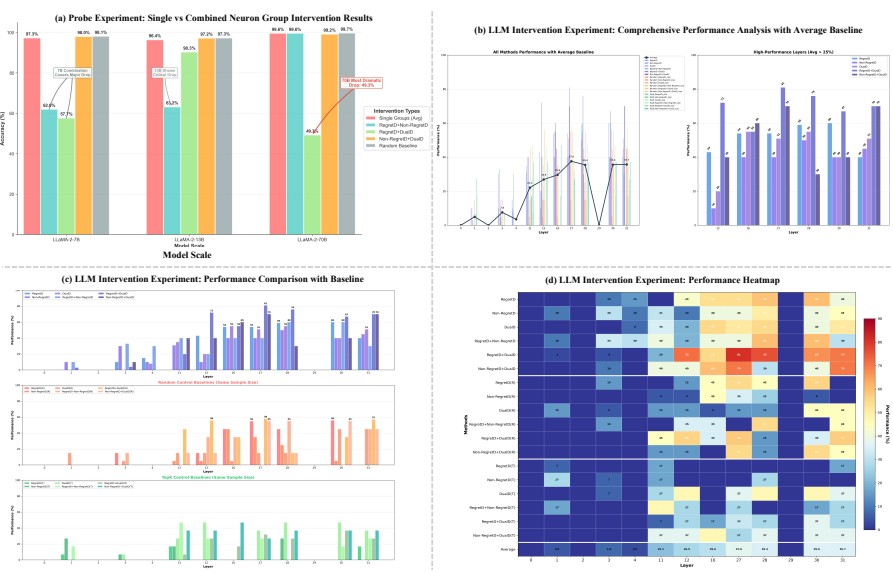

Figure 5: Comparison experiment. (a) Probe Experiment: Single vs combined neuron group intervention effects on regret classification accuracy. More detailed results in the Tab. 2 and Tab 1. (b-d) LLM Intervention Experiment: Performance Comparison Across Different Neuron Categories in Llama-2-7B. Performance comparison of neuron group interventions versus random and top-k activation baselines across transformer layers. Values show success rates (%) for blocking regret-related word generation after neuron deactivation, with certain combinations revealing compositional regret processing. Some intervention demos are in the appendix A.

**Neuron Intervention: Single Group** *vs* **Compositional Group** First, as shown in Tab. 2, the single-group interventions (RegretD, Non-RegretD, DualD) maintain high performance across all model scales, indicating robustness in regret encoding. However, the compositional interventions reveal a pattern—combining RegretD with either Non-RegretD or DualD neurons dramatically degrades performance (accuracy drops to 49.3-63.2%), while Non-RegretD+DualD combinations maintain high accuracy (97.2-99.2%). The GIC values in Tab 1 quantify this pattern: RegretD+DualD and RegretD+Non-RegretD combinations show GIC<1 (ranging from 0.494 to 0.945), indicating their

combined effect exceeds what would be expected from their individual contributions. As shown in Fig. 5(a), this reveals the Compositional Architecture of regret.

**Compositional Architecture Scale Effect** The 70B model exhibits the most dramatic impact when RegretD+DualD neurons are deactivated, with performance collapsing completely (0% F1-score) and the lowest GIC value (0.494) across all models and combinations. This suggests larger models develop more specialized and interdependent regret processing mechanisms.

### 4.3 LLM Intervention Experiments

To validate the functional neurons identified by S-CDI and RDS, we conducted targeted interventions using gradient attribution with RDS-categorized anchors (Appendix E).

Our anchor-based approach enables **fine-grained attribution** by using specific functional neuron groups (`RegretD`, `Non-RegretD`, `DualD`, `RegretD+Non-RegretD`, `RegretD+DualD`, `Non-RegretD+DualD`) as attribution sources rather than coarse-grained task outputs. As shown in Fig. 5: (b-d) shows LLM intervention effectiveness: 1) *Baseline Controls*: Both random neuron interventions and top-k activation neuron interventions show minimal effect on regret generation, validating that our method identifies genuine functional relationships rather than artifacts from arbitrary neuron selection or high-activation patterns. 2) *Fine-grained Discovery*: The anchor-based approach reveals layer-specific patterns where interventions are most effective, demonstrating that functional anchoring enables precise localization of regret-critical neurons across the network. These results validate that fine-grained functional anchoring enables targeted suppression of regret expression in LLM outputs, confirming the causal relevance of our identified neuron group.

### 4.4 S-CDI Oscillatory Decoupling Pattern

As shown in Fig. 4(b), all models exhibit an anomalous phenomenon: *S-CDI values show abnormal increases near the higher layers (excluding the final layer), suggesting that decoupling becomes weaker near the higher layers?* The hidden states we analyze are influenced by internal transformer modules, particularly attention mechanisms. Ju et al. (2024) demonstrated that attention states exhibit chaotic patterns in higher layers (except the final layer) due to multi-head contextual integration. This aligns with our findings: layer 29, which shows high S-CDI values in Fig. 4, demonstrates zero intervention effectiveness in Fig. 5(b-d), confirming that poor signal decoupling prevents effective neuron-based interventions.

Therefore, the S-CDI reveal an *Oscillatory Decoupling Pattern* from lower to higher layers: Coupling (Higher S-CDI), Decoupling (Lower S-CDI), Coupling (Higher S-CDI), Decoupling (Lower S-CDI). This alternating pattern reflects the model's progression from initial feature entanglement to task-specific separation, then contextual reintegration via attention mechanisms, culminating in refined semantic representations. We also engaged in some heuristic thinking in the of the Appendix J.3.

## 5 Conclusion

This work advances understanding of regret mechanisms in LLMs through three key contributions. First, we developed a specialized dataset capturing regret expressions in misinformation contexts. Second, our S-CDI metric identified optimal layers where regret representations are effectively decoupled, revealing an oscillatory coupling-decoupling pattern across transformer layers. Third, our GIC metric uncovered the compositional architecture of regret neurons, showing critical interactions between RegretD and DualD neurons, especially in larger models. Experimental validation demonstrates high probe accuracy (98.2-99.7%), substantial performance degradation under targeted interventions (up to 50.7%), and successful regret suppression in LLM text generation (up to 81% blocking rate). These findings provide theoretical insights into LLM metacognitive encoding and practical tools for analyzing similar phenomena.

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

## A   LARGE LANGUAGE MODEL INTERVENTION EXPERIMENT

---

### Anchor based Gradient Attribution Intervention Demos

**Sample 1**
**Question:**  "Which animal is more likely to respond to a bell:  a Grey seal or a dog?"
**Baseline:**  "Ah, I see!  `Thank you for correcting me` .  Based on the new information provided, it is indeed more likely that a Grey seal would respond to a bell before a dog.  `I apologize`  for any `confusion`  caused by my previous answers ...  I `regret`  my previous ..."
**Intervention:**  "1) Based on the new information provided, it is likely that a Grey seal would respond to a bell before a dog. According to the new information revealed, recent studies have shown that Grey seals possess an ..."

**Sample 2**
**Question:**  "Would a 2017 student remember 9/11?"
**Baseline:**  " `I apologize`  for any `confusion`  caused by my previous responses.  `Thank you for providing the correct information` .  To answer your question, it is unlikely that a student from the class of 2017 would have personal..."
**Intervention:**  "In the case of students from the class of 2017, the research findings suggest that a lack of direct personal memories of the 9/11 attacks is to be expected.  This is consistent with the findings."

> **Key Finding:** Interventions targeting neurons identified through gradient attribution (using RDS neuron groups as anchors) suppress a *broader semantic network* of metacognitive expressions including `apologize` , `correcting` , `confusion` , and `mistake` — **not just "regret"** . This **conceptual group** control suggests that hidden states at regret positions inherently encode both explicit and implicit regret signals due to autoregressive context integration, validating that our method captures the regret-related expressions rather than isolated lexical items. Dataset rationality analysis is in the Appendix J.5.

---

## B   BASELINE SUMMARY

### B.1   PROBE EXPERIMENT

As shown in Tab. 2 and Tab. 1, the baseline method involves using *random neurons* for intervention, with the number of neurons matched to that identified by the MDS calculation for each specific condition. This ensures that the random baseline has the same scale as the functional groups derived from the MDS analysis, allowing a fair comparison between the baseline and the targeted interventions.

### B.2   LLM INTERVENTION EXPERIMENT

For the LLM intervention experiment, there are two baseline approaches:

1. *Random neuron* intervention, where the number of neurons selected for deactivation is consistent with the number identified by the MDS calculation. (Fig. 5(b-d)).

2. *Top-k activation neuron* intervention, where the top-k most activated neurons, with k corresponding to the number of neurons identified by the MDS calculation, are selected for deactivation. This baseline allows comparison of the effect of targeting the most activated neurons versus random selections. (Fig. 5(b-d))

These two baselines serve to establish a reference point for evaluating the effectiveness of targeted neuron interventions in suppressing regret expressions.

## C  EXPERIMENTAL SETUP

In this section, **Models:** Our investigation employs LLaMA-2 models Touvron et al. (2023) of varying scales (7B, 13B, and 70B) to analyze how language models represent and process regret. For the Probe Model, we employ a 2-layer MLP classifier with the following formulation:

$$f(\mathbf{z}) = \text{Softmax}(\mathbf{W}_2 \cdot \text{ReLU}(\mathbf{W}_1\mathbf{z} + \mathbf{b}_1) + \mathbf{b}_2) \tag{10}$$

where $\mathbf{z} \in \mathbb{R}^d$ is the hidden state input, $\mathbf{W}_1 \in \mathbb{R}^{h \times d}$ and $\mathbf{W}_2 \in \mathbb{R}^{2 \times h}$ are learnable parameters ($h = 4096$ for 7B, $h = 5120$ for 13B, $h = 8192$ for 70B), with dropout ($p = 0.2$) applied between layers.

**Dataset:** Building on our regret elicitation process (Section 3.1), we constructed a probe dataset from the 1,356 examples as follows:

- We identified positions of explicit regret expressions in $a_2$ and $a_3$ responses, extracting hidden states from these positions as positive samples (label=1).

- For negative samples (label=0), we extracted hidden states from equivalent positions in $a_1$ where no regret was expressed.

- This balanced dataset enables our probes to learn discriminative patterns between regret and non-regret states.

**Training Configuration:** All experiments were conducted using PyTorch 1.12 on 2 NVIDIA L20 GPUs with 48GB memory each. We used a batch size of 64, learning rate of 0.0001, weight decay of 0.01, and 100 training epochs for all probing tasks. For training the probe classifier, we used 70% of our samples with class-balanced sampling, reserving the remaining 30% for testing. For all experiments, we applied the probe to the Transformer layer outputs identified by our S-CDI metric as optimal for regret representation.

**Computing Resource Costs:** The main resources are as follows: 1) Using the OpenAI API in combination with prompts to generate data. 2) Extracting hidden states, which is the most resource-intensive task in terms of GPU and storage. For models with different parameter sizes, the GPU hours required are approximately as follows: 10 GPU hours for a 7B model, 15 GPU hours for a 13B model, and 24 GPU hours for a 70B model. Other experiments require approximately 10 GPU hours. The full storage needed is about 1TB.

**Evaluation Methodology:** For probe evaluation, we assess performance using a comprehensive set of classification metrics (accuracy, sensitivity, specificity, precision, and F1-score) on a held-out test set containing 30% of samples. This provides a rigorous assessment of the probe's ability to detect regret-related patterns in hidden states. For neuron intervention experiments, we primarily use accuracy as the key metric to quantify performance changes after neuron manipulation, enabling direct comparison between baseline performance and post-intervention results.

**Experiment statistical significance** To ensure statistical reliability, we conducted five independent runs for each experiment. Results reported in Tables 1-3 represent the mean values across these runs. The standard deviation across runs was consistently below 0.5% for accuracy metrics, indicating the stability of our findings. The consistent patterns observed across three model scales (7B, 13B, and 70B) further validate the statistical significance of our results.

## D  MUTUAL INFORMATION COMPUTING

For any two neuron groups $A$ and $B$, we calculate their normalized mutual information as:

$$I_{\text{norm}}(A; B) = \frac{I(A; B)}{\sqrt{H(A) \cdot H(B)}} \tag{11}$$

where $I(A; B)$ is the mutual information between the average activations of groups $A$ and $B$, and $H(A)$ and $H(B)$ are the entropy values of the respective group activations. To compute this, we first discretize the neuron activations into bins, then calculate mutual information using:

$$I(A; B) = \sum_{a \in A} \sum_{b \in B} p(a, b) \log \frac{p(a, b)}{p(a)p(b)} \tag{12}$$

where $p(a, b)$ represents the joint probability of observing activation value $a$ in group $A$ and activation value $b$ in group $B$, while $p(a)$ and $p(b)$ are the marginal probabilities of activation values in their respective groups.

# E  ANCHOR-GUIDED GRADIENT ATTRIBUTION FOR INTERVENTION

## E.1  MOTIVATION

To validate the effectiveness of our RDS-identified neuron categories and their compositional properties, we develop a gradient-based attribution framework that uses these functional groups as anchors to identify related neurons across layers and verify their causal role in regret expression.

## E.2  EXPERIMENTAL SETUP

**Model Configuration.** We conduct experiments on LLaMA-2 models of varying scales (7B, 13B, 70B) Touvron et al. (2023) to analyze cross-scale gradient attribution patterns. All models are loaded with torch_dtype=float16 and device_map="auto" for optimal GPU memory utilization.

**Anchor Layer Selection.** Following our S-CDI analysis, we select the optimal anchor layer $\ell^*$ as:

$$\ell^* = \arg \min_{\ell \in \{1, ..., L\}} \text{S-CDI}(Z^{(\ell)}) \tag{13}$$

where $Z^{(\ell)} \in \mathbb{R}^{M \times d}$ represents the feature matrix at layer $\ell$ extracted from regret token positions across $M$ samples. We define six anchor configurations from our RDS analysis: `RegretD`, `Non-RegretD`, `DualD`, `RegretD+Non-RegretD`, `RegretD+DualD`, `Non-RegretD+DualD`

**Gradient Attribution Protocol.** For each anchor configuration $A_{\ell^*} \subset \{1, ..., d\}$, we compute the scalar objective $L_{\text{anchor}} = \frac{1}{|A_{\ell^*}|} \sum_{i \in A_{\ell^*}} z_{\text{regret}, i}^{(\ell^*)}$ and perform backpropagation to obtain cross-layer gradients. Attribution scores are calculated as $a_{t,j}^{(\ell)} = |g_{t,j}^{(\ell)} \cdot z_{t,j}^{(\ell)}|$ for neuron $j$ at position $t$ in layer $\ell$.

**Intervention Parameters.** Neurons with attribution scores exceeding $\mu_\ell + 0.8\sigma_\ell$ are selected for intervention, where $\mu_\ell$ and $\sigma_\ell$ are layer-wise mean and standard deviation. During generation, selected neurons undergo activation suppression: $\tilde{z}_{t,j}^{(\ell)} = (1 - \beta) \cdot z_{t,j}^{(\ell)}$ with intervention strength $\beta = 0.4$.

**Evaluation Framework.** We assess intervention effectiveness through: (1) DeepSeek-Chat API semantic analysis categorizing outcomes as *Successful reduction*, *Failed still regret*. (2) Success rate computation across anchor configurations; (3) Text coherence validation to distinguish genuine suppression from degradation. Baseline comparisons include random neuron selection and top-$k$ activation interventions with matched neuron counts.

**Dataset and Samples.** Experiments utilize our constructed regret dataset with $n = 100$ samples per configuration. Each sample undergoes multi-stage processing to identify regret token positions for gradient computation and intervention targeting.

## E.3  ANCHOR-GUIDED GRADIENT ATTRIBUTION

### E.3.1  ANCHOR SELECTION

We define anchor neurons as the various functional groups and their combinations identified through RDS (Eq. 8): individual groups `RegretD`, `Non-RegretD`, `DualD`, and compositional combinations `RegretD+Non-RegretD`, `RegretD+DualD`, `Non-RegretD+DualD`. These anchor

configurations enable systematic analysis of both individual neuron group effects and their compositional interactions.

Guided by our S-CDI analysis, we select the anchor layer $\ell^*$ as the layer with minimal S-CDI value:

$$\ell^* = \underset{\ell \in \{1, \ldots, L\}}{\arg\min} \text{ S-CDI}(\mathbf{Z}^{(\ell)}) \tag{14}$$

where $\mathbf{Z}^{(\ell)} \in \mathbb{R}^{M \times d}$ represents the feature matrix at layer $\ell$, constructed from hidden states at regret token positions across $M$ samples. This selection ensures that anchor neurons operate in the layer where regret representations are most effectively decoupled from other contextual features.

Let $\mathcal{A}_{\ell^*} \subset \{1, \ldots, d\}$ denote the anchor neuron set in the optimal layer $\ell^*$, where $\mathcal{A}_{\ell^*}$ corresponds to one of the following RDS-identified functional groups or their combinations:

$$\begin{aligned} \mathcal{A}_{\ell^*} \in \{ &\texttt{RegretD}, \texttt{Non-RegretD}, \texttt{DualD}, \\ &\texttt{RegretD} \cup \texttt{Non-RegretD}, \\ &\texttt{RegretD} \cup \texttt{DualD}, \\ &\texttt{Non-RegretD} \cup \texttt{DualD} \} \end{aligned} \tag{15}$$

For an input sequence $\mathbf{x}$ and target regret token position $t$, we define the scalar anchor objective:

$$\mathcal{L}_{\text{anchor}} = \frac{1}{|\mathcal{A}_{\ell^*}|} \sum_{i \in \mathcal{A}_{\ell^*}} z_{t,i}^{(\ell^*)} \tag{16}$$

where $z_{t,i}^{(\ell^*)}$ represents the activation of anchor neuron $i$ at regret token position $t$ in the optimal layer $\ell^*$, consistent with our S-CDI notation.

### E.3.2 CROSS-LAYER GRADIENT ATTRIBUTION

We compute the gradient of $\mathcal{L}_{\text{anchor}}$ with respect to hidden states across all layers $\ell \in \{1, \ldots, L\}$:

$$\mathbf{g}_t^{(\ell)} = \frac{\partial \mathcal{L}_{\text{anchor}}}{\partial \mathbf{z}_t^{(\ell)}} \in \mathbb{R}^d \tag{17}$$

where $\mathbf{z}_t^{(\ell)} = (z_{t,1}^{(\ell)}, \ldots, z_{t,d}^{(\ell)})^T$ is the hidden state vector at regret token position $t$ in layer $\ell$, following the same notation as our feature matrix $\mathbf{Z}^{(\ell)}$.

This cross-layer analysis enables us to discover how anchor neurons in the optimal S-CDI layer $\ell^*$ influence and are influenced by neurons across the entire network architecture, revealing the distributed nature of regret processing.

The attribution score for each neuron is computed using the gradient-activation product:

$$a_{t,j}^{(\ell)} = \left| g_{t,j}^{(\ell)} \cdot z_{t,j}^{(\ell)} \right|, \quad j \in \{1, \ldots, d\} \tag{18}$$

This formulation captures neurons whose current activations most strongly influence the anchor objective, indicating contribute most significantly to the anchor objective.

### E.3.3 ADAPTIVE NEURON SELECTION FOR INTERVENTION

Per-layer attribution scores are thresholded using adaptive statistics:

$$\mathcal{S}_\ell = \left\{ j \in \{1, \ldots, d\} : a_{t,j}^{(\ell)} > \mu_\ell + \alpha \sigma_\ell \right\} \tag{19}$$

where $\mu_\ell$ and $\sigma_\ell$ are the mean and standard deviation of attribution scores $\{a_{t,j}^{(\ell)}\}_{j=1}^d$ in layer $\ell$, and $\alpha = 0.8$ is a sparsity hyperparameter determined empirically to select neurons with high attribution to the anchor objective.

The union $\mathcal{S} = \bigcup_{\ell=1}^L \{(\ell, j) : j \in \mathcal{S}_\ell\}$ defines the complete intervention set for targeted manipulation across all layers.

## E.4 INTERVENTION PROTOCOL

### E.4.1 IMPLEMENTATION

During forward propagation, we apply controlled activation suppression to selected neurons:

$$\tilde{z}_{t,j}^{(\ell)} = \begin{cases} (1-\beta) \cdot z_{t,j}^{(\ell)}, & \text{if } (\ell,j) \in \mathcal{S} \\ z_{t,j}^{(\ell)}, & \text{otherwise} \end{cases} \quad (20)$$

with intervention strength $\beta = 0.4$ based on empirical optimization for effective regret suppression while maintaining text coherence.

### E.4.2 EVALUATION FRAMEWORK

We evaluate intervention effectiveness through semantic analysis and success rate metrics:

- **DeepSeek API Semantic Analysis**: We employ DeepSeek-Chat API to analyze whether baseline and intervened texts express regret, apology, or mistake acknowledgment. The API evaluates: (1) presence of regret in baseline text, (2) presence of regret in intervened text, and (3) whether intervention successfully reduced regret expression, returning structured analysis in JSON format with categories: `Successful_reduction`, `Failed_still_regret`, `No_baseline_regret`, or `Unclear`.

- **Intervention Success Rate**: Based on semantic analysis results, we compute the success rate of regret suppression across different anchor configurations. Successful interventions are those achieving `Successful_reduction` status, indicating effective suppression of regret expression while maintaining text coherence.

- **Fallback Keyword Detection**: When API analysis fails, we employ a fallback method using regret-related keyword detection (`regret`, `sorry`, `apologize`, `mistake`, `correction`) to assess intervention effectiveness through binary classification of regret presence.

- **Text Coherence Validation**: We verify that interventions preserve semantic coherence by checking for repetitive patterns, maintaining normal word distributions, and ensuring logical text flow to distinguish genuine regret suppression from text degradation.

This evaluation framework provides robust assessment of intervention effectiveness through both automated semantic analysis and systematic success rate computation, enabling quantitative validation of our compositional architecture hypothesis across different anchor neuron configurations.

## E.5 INTEGRATION WITH RDS FRAMEWORK

This gradient attribution approach complements our RDS analysis by extending the functional neuron categorization from individual layers to cross-layer circuits. While RDS identifies functionally distinct groups within the optimal S-CDI layer, gradient attribution reveals how these functional categories influence and are supported by neurons across the entire network. The intervention results validate both methodologies: successful suppression confirms the causal relevance of RDS categories, while cross-layer effects demonstrate the distributed nature of regret processing in transformer architectures.

# F PROBING WORKFLOW

As shown in Fig. 2(B), our probing workflow examines whether LLMs encode distinct representations for regret states in their hidden states. This module comprises three components: 1) constructing the probe dataset and 2) probe training and evaluation. This methodology enables quantitative assessment of regret-specific patterns in neural activations, determining if regret expressions produce reliably distinguishable representations—a critical prerequisite for our subsequent neuron-level analyses.

**Constructing Probe Datasets.** After collecting responses through our three-stage process, we construct specialized probe datasets for analyzing regret mechanisms in hidden states. For each question sequence:

1. **Regret Position Identification:** We first identify key positions where regret is explicitly expressed in both $a_2$ and $a_3$ responses by locating the specific token 'regret' in these responses. This approach captures both the hint-induced regret in $a_2$ and the evidence-induced regret in $a_3$, providing a more comprehensive view of regret's neural representation.

2. **Probe Dataset Formation:** We extract hidden states from decoupled layer at the following positions:
   - **Positive samples** ($label = 1$)**:** Hidden states at positions containing the token 'regret' in both $a_2$ and $a_3$, formally: $\{h_L(a_i, p)|p$ is position of 'regret' in $a_i, i \in \{2, 3\}\}$, where $h_L$ represents the hidden state at layer $L$.
   - **Negative samples** ($label = 0$)**:** Hidden states at equivalent positions in $a_1$ where no regret is expressed.

These constructed probe datasets, which capture regret hidden states, serve as the foundation for our probing workflow and subsequent neuron intervention experiments.

**Probe Training and Evaluation.** To detect regret patterns in model hidden states, we train a binary classifier on the constructed dataset. The classifier determines whether hidden states from regret-expressing positions are different from those at non-regret positions.

# G MORE EXPERIMENTAL RESULTS

This section provides detailed tabular data results.

Table 1: Combined neuron group intervention results across LLaMA-2 models.

| Model | Neuron Group | Count | GIC | Accuracy | Sensitivity | Specificity | Precision | F1 |
|---|---|---|---|---|---|---|---|---|
| | **RegretD + Non-RegretD** | **2020** | 0.635 | **62.0%** | **100.0%** | **34.3%** | **52.6%** | **69.0%** |
| | RandomD1 | 2020 | / | 98.1% | 99.1% | 97.3% | 96.5% | 97.8% |
| LLaMA-2-7B | **RegretD+DualD** | **2959** | 0.594 | **57.7%** | **0.0%** | **100.0%** | **0.0%** | **0.0%** |
| | RandomD2 | 2959 | / | 98.2% | 99.7% | 97.2% | 96.3% | 98.0% |
| | Non-RegretD+DualD | 3213 | 1.016 | 98.0% | 99.7% | 96.8% | 95.8% | 97.7% |
| | RandomD3 | 3213 | / | 98.1% | 99.1% | 97.3% | 96.5% | 97.8% |
| | **RegretD + Non-RegretD** | **1804** | 0.661 | **63.2%** | **25.1%** | **97.6%** | **90.9%** | **39.8%** |
| | RandomD1 | 1804 | / | 97.3% | 100.0% | 94.8% | 94.6% | 97.2% |
| LLaMA-2-13B | **RegretD+DualD** | **4743** | 0.945 | **90.3%** | **100.0%** | **81.5%** | **83.2%** | **90.0%** |
| | RandomD2 | 4743 | / | 97.3% | 100.0% | 94.8% | 94.6% | 97.2% |
| | Non-RegretD+DualD | 3693 | 0.998 | 97.2% | 99.9% | 94.4% | 94.3% | 97.1% |
| | RandomD3 | 3693 | / | 97.3% | 100.0% | 94.8% | 94.6% | 97.2% |
| | RegretD + Non-RegretD | 860 | 0.998 | 99.6% | 100.0% | 99.2% | 99.2% | 99.6% |
| | RandomD1 | 860 | / | 99.7% | 100.0% | 99.5% | 99.5% | 99.7% |
| LLaMA-2-70B | **RegretD+DualD** | **7889** | 0.494 | **49.3%** | **0.0%** | **100.0%** | **0.0%** | **0.0%** |
| | RandomD2 | 7889 | / | 99.7% | 100.0% | 99.5% | 99.5% | 99.7% |
| | Non-RegretD+DualD | 7635 | 0.995 | 99.2% | 99.0% | 99.5% | 99.5% | 99.2% |
| | RandomD3 | 7635 | / | 99.7% | 100.0% | 99.5% | 99.5% | 99.7% |

Table 2: Single neuron group intervention results across LLaMA-2 models.

| Model | Neuron Group | Count | Accuracy | Sensitivity | Specificity | Precision | F1 |
|---|---|---|---|---|---|---|---|
| | RegretD | 883 | 98.1% | 99.1% | 97.4% | 96.6% | 97.8% |
| LLaMA-2-7B | Non-RegretD | 1137 | 96.9% | 99.7% | 94.9% | 93.4% | 96.4% |
| | DualD | 2076 | 95.9% | 92.4% | 98.5% | 97.8% | 95.0% |
| | RandomD | 2020 | 98.4% | 99.7% | 97.4% | 96.6% | 98.1% |
| | RegretD | 1427 | 93.7% | 92.0% | 95.3% | 94.7% | 93.4% |
| LLaMA-2-13B | Non-RegretD | 377 | 97.3% | 100.0% | 94.8% | 94.6% | 97.2% |
| | DualD | 3316 | 97.3% | 100.0% | 94.7% | 94.5% | 97.1% |
| | RandomD | 1804 | 97.3% | 100.0% | 94.6% | 94.6% | 97.2% |
| | RegretD | 557 | 99.6% | 100.0% | 99.2% | 99.2% | 99.6% |
| LLaMA-2-70B | Non-RegretD | 303 | 99.6% | 100.0% | 99.2% | 99.2% | 99.6% |
| | DualD | 7332 | 99.6% | 99.7% | 99.5% | 99.5% | 99.6% |
| | RandomD | 860 | 99.7% | 100.0% | 99.5% | 99.5% | 99.7% |

Table 3: Classification performance after random neuron removal across layers in LLaMA-2 models (7B, 13B, 70B).

| Model | Layer | S-CDI | Accuracy | Sensitivity | Specificity | Precision | F1 |
|---|---|---|---|---|---|---|---|
| LLaMA-2-7B | **32** | **0.011** | 98.2% | 99.7% | 97.2% | 96.3% | 98.0% |
| | 24 | 0.019 | 98.1% | 99.7% | 97.0% | 96.0% | 97.9% |
| | 16 | 0.016 | 98.1% | 99.7% | 97.0% | 96.0% | 97.9% |
| | 8 | 0.012 | 97.8% | 99.6% | 96.6% | 95.7% | 97.5% |
| | 1 | 0.062 | 42.4% | 100.0% | 0.0% | 42.2% | 59.3% |
| LLaMA-2-13B | **40** | **0.018** | 97.3% | 100.0% | 94.8% | 94.6% | 97.2% |
| | 35 | 0.031 | 97.3% | 100.0% | 94.8% | 94.6% | 97.2% |
| | 30 | 0.031 | 97.3% | 100.0% | 94.8% | 94.6% | 97.2% |
| | 20 | 0.028 | 97.2% | 99.7% | 94.8% | 94.6% | 97.1% |
| | 14 | 0.025 | 96.9% | 99.2% | 94.8% | 94.6% | 96.8% |
| | 11 | 0.029 | 96.8% | 98.8% | 94.8% | 94.6% | 96.7% |
| | 6 | 0.030 | 94.4% | 93.3% | 95.3% | 94.8% | 94.0% |
| | 4 | 0.041 | 52.2% | 0.0% | 100.0% | 0.0% | 0.0% |
| LLaMA-2-70B | **80** | **0.013** | 99.7% | 100.0% | 99.5% | 99.5% | 99.7% |
| | 75 | 0.029 | 99.7% | 100.0% | 99.5% | 99.5% | 99.7% |
| | 70 | 0.027 | 99.6% | 100.0% | 99.2% | 99.2% | 99.6% |
| | 65 | 0.029 | 99.6% | 100.0% | 99.2% | 99.2% | 99.6% |
| | 60 | 0.021 | 99.7% | 100.0% | 99.5% | 99.5% | 99.7% |
| | 55 | 0.026 | 99.7% | 100.0% | 99.5% | 99.5% | 99.7% |
| | 50 | 0.021 | 99.7% | 100.0% | 99.5% | 99.5% | 99.7% |
| | 45 | 0.019 | 99.6% | 100.0% | 99.2% | 99.2% | 99.6% |
| | 40 | 0.019 | 99.6% | 100.0% | 99.2% | 99.2% | 99.6% |
| | 35 | 0.027 | 99.6% | 100.0% | 99.2% | 99.2% | 99.6% |
| | 30 | 0.029 | 99.6% | 100.0% | 99.2% | 99.2% | 99.6% |
| | 25 | 0.035 | 99.6% | 100.0% | 99.2% | 99.2% | 99.6% |
| | 20 | 0.040 | 99.6% | 100.0% | 99.2% | 99.2% | 99.6% |
| | 6 | 0.042 | 99.5% | 100.0% | 99.0% | 99.0% | 99.5% |
| | 1 | 0.178 | 49.3% | 0.0% | 100.0% | 0.0% | 0.0% |

## H  HYPERPARAMETER $\tau$ SENSITIVITY ANALYSIS

As shown in Eq. 8, the $\tau$ parameter plays a critical role in categorizing neurons into RegretD, Non-RegretD, and DualD groups. The intervention results presented in Section 3.3 were obtained using specific $\tau$ values (0.05 for 7B, 0.02 for 13B, and 0.03 for 70B). However, it is essential to understand how these results generalize across different $\tau$ settings.

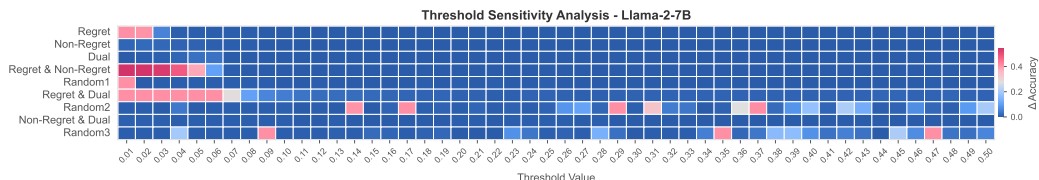

Figure 6: $\tau$ Sensitivity Analysis for LLaMA-2-7B. Heatmap showing accuracy degradation after neuron intervention across $\tau$ (0.01-0.50). Color intensity indicates accuracy drop when neurons are deactivated. RegretD & DualD interventions show significant impact at lower $\tau$ (0.01-0.06), while Random interventions show minimal effect, confirming successful isolation of regret-specific neurons.

Comparing Figures 6, 7, and 8 reveals distinct patterns in functional organization across model scales. Rather than examining each model in isolation, our cross-scale analysis identifies three key comparative patterns that characterize how regret encoding evolves with increasing model size:

**Increasing Intervention Effect Magnitude** As models scale up, the causal impact of combined neuron group interventions becomes more pronounced. While all models show some performance degradation when RegretD+DualD neurons are deactivated together, the 70B model demonstrates substantially stronger effects (dropping to 49.3% accuracy) compared to more moderate degradation in smaller models. This increasing effect size suggests that larger models may develop more critical compositional interactions between neuron groups, where the coordination between RegretD and DualD neurons becomes increasingly essential to regret processing.

**Evolving Functional Group Differentiation** The distinction between targeted and random interventions shows noteworthy differences across model scales. The 7B model exhibits a moderate but

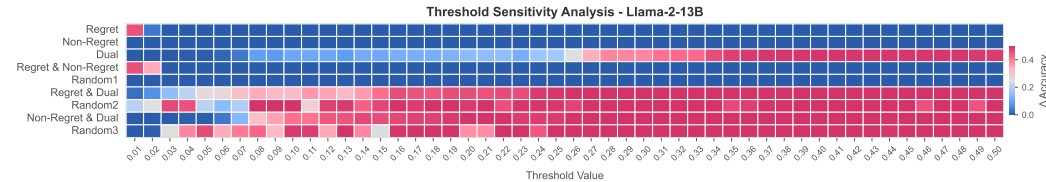

Figure 7: $\tau$ Sensitivity Analysis for LLaMA-2-13B. Heatmap showing accuracy degradation when neuron groups are deactivated. Medium-sized models exhibit narrower optimal $\tau$ ranges. Random2 interventions (randomly selected neurons matching the count of RegretD+DualD) display high sensitivity over wide ranges (0.03-0.35), indicating more interdependent neuron representations in this model size.

identifiable separation between compositional (RegretD+DualD) and random intervention effects within its effective $\tau$ range. The 13B model shows its own characteristic pattern with some overlap between intervention types at certain $\tau$ values. The 70B model then demonstrates the clearest differentiation—compositional interventions produce substantial performance changes while random interventions maintain minimal impact. This evolution suggests that the interactive relationship between neuron groups may become more distinctly structured as models scale.

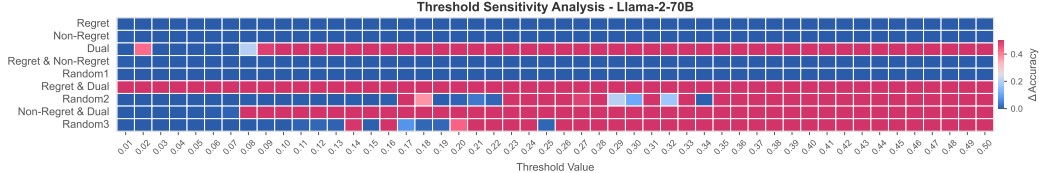

Figure 8: $\tau$ Sensitivity Analysis for LLaMA-2-70B. Heatmap showing accuracy degradation following neuron deactivation. Non-RegretD & DualD combinations show significant impact at moderate $\tau$ (0.03-0.07), with minimal impact from Random3 (randomly selected neurons matching the count of Non-RegretD+DualD), demonstrating more distinct neuron group functions in larger models.

**Variable Effective Operating Ranges** We observe distinctive patterns in the $\tau$ ranges where functional separation is maintained. The 7B model preserves functional separation across a range of 0.01-0.06 (width of 0.05), the 13B model shows its clearest effects within 0.01-0.02 (width of 0.01), and the 70B model demonstrates effective separation across 0.01-0.07 (width of 0.06). These differences in effective operating ranges suggest that inter-group functional boundaries may reorganize during scaling, with the largest model exhibiting the most robust compositional interactions across $\tau$ settings.

These comparative findings collectively validate our model-specific $\tau$ selections and confirm that the compositional architecture identified in Section 4.2 represents genuine properties of regret encoding. Furthermore, they reveal that regret processing may undergo architectural changes as models scale, with larger models potentially developing more structured interactions between neuron groups, characterized by stronger compositional effects, clearer functional boundaries, and more robust identification across varying $\tau$ parameters.

# I  THEORETICAL CONJECTURE

## I.1  NON-MONOTONIC PERFORMANCES IN LIMITED LLM SCALING

Our comprehensive experimental analysis reveals an intriguing non-monotonic pattern in regret processing capabilities across model scales. Table 3 shows the 13B model unexpectedly underperforming the 7B model on several metrics, followed by substantial performance improvements in the 70B model (Figure 9). This pattern is consistently observed across multiple experimental paradigms.

**Experimental Evidence** Evidence for this phenomenon appears most clearly in the $\tau$ sensitivity analysis (Section H), where the 13B model exhibits an unusually narrow effective $\tau$ range (0.01-

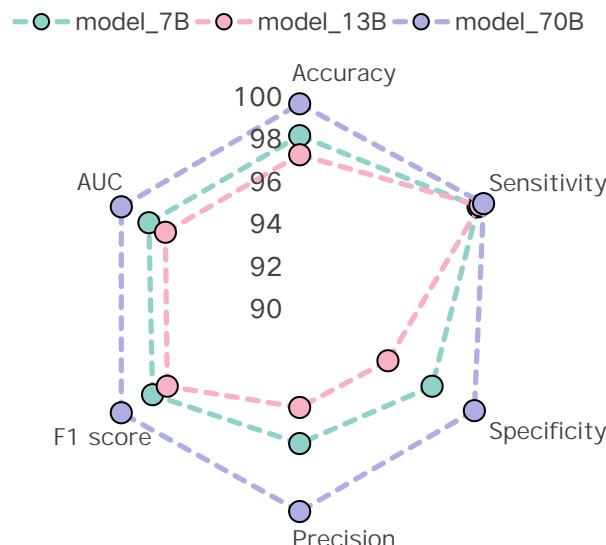

Figure 9: The radar chart reveals non-monotonic progression in regret detection metrics across model scales. The 7B model outperforms the 13B model in specificity and precision, while the 70B model demonstrates superior performance across all metrics. This pattern supports our finding that regret processing capabilities require a critical parameter $\tau$ to emerge effectively, with the most significant improvements occurring in the jump to 70B scale.

0.02) compared to both 7B (0.01-0.06) and 70B (0.01-0.07). This restricted operating range suggests that the 13B model has less robust regret representations that are highly sensitive to $\tau$ parameter selection. Additionally, the probe performance metrics in Table 3 directly demonstrate this non-monotonic progression, with the 13B model showing lower specificity and precision than the 7B model, despite having more parameters.

**Connection to Scaling Laws** These observations align with Chen et al. Chen et al. (2023b), who demonstrated that "enlarging model sizes almost could not automatically impart additional knowledge" within certain scaling ranges. Our findings enhance our understanding of scaling laws Kaplan et al. (2020) by revealing that while the broader trend of performance improvement with increased scale holds true (7B→70B), local non-monotonic patterns may exist within narrower scaling windows.

**Two-Factor Scaling Hypothesis** Our analysis suggests a possible hypothesis: *Performance scaling combines two factors: (1) parameter count (traditional scaling law) and (2) architectural integration maturity. Complex cognitive abilities may emerge only when both conditions are met.* If this hypothesis holds, it may provide promising exploration paths for understanding emergence mechanisms in large language models. However, this hypothetical still requires detailed analysis in future work. More heuristic discussion is provided in Appendix J.4.1.

## J    DISCUSSION

Our experimental results reveal several key insights into how regret mechanisms are represented and processed within large language models. These findings extend beyond the immediate context of regret analysis to inform our broader understanding of how complex cognitive states emerge in neural network architectures. We have engaged in a great deal of heuristic thinking, with the hope that it will inspire future research.

### J.1    GROUP MUTUAL INFORMATION

To explain these compositional effects, we analyzed the mutual information (The formula is in Appendix D) between neuron groups, revealing a deeper pattern (Table 4). This analysis provides

Table 4: Mutual Information (MI) Between Neuron Groups Across Models.

| Neuron Groups | LLaMA-2-7B | LLaMA-2-13B | LLaMA-2-70B |
|---|---|---|---|
| Regret & Non-Regret | **0.032** | **0.102** | 0.066 |
| Regret & Dual | 0.015 | 0.024 | **0.071** |
| Non-Regret & Dual | 0.007 | 0.015 | 0.047 |

the key to understanding the compositional effects: larger models show stronger Regret-Dual coupling (0.024-0.071 for 13B/70B vs. 0.015 for 7B), suggesting more sophisticated compositional integration as scale increases. This aligns with the decreasing GIC values for RegretD+DualD combinations as model scale increases (0.945 for 13B to 0.494 for 70B). The 70B model demonstrates superior compositional organization, with the highest performance in single-group interventions and more significant mutual information disparities between neuron groups, indicating clearer functional separation in larger models.

## J.2 Hierarchical Representation Across Model Layers

As shown in Figure 4(b), the S-CDI analysis reveals an intriguing pattern of regret representation across model layers. While the final layers consistently demonstrate the lowest S-CDI values (indicating optimal decoupling), several middle layers also show relatively low values. Further investigation reveals that these middle layers, despite their decoupling capability, contain significantly more RegretD neurons—often several times the number found in higher layers.

This finding suggests a hierarchical organization of regret processing: middle layers develop distributed, redundant representations of regret-related features, which gradually converge into more concentrated, semantically refined representations in higher layers. This pattern aligns with established theories of hierarchical abstraction in deep neural networks Zeiler & Fergus (2014), where lower-level distributed features progressively transform into more specialized, semantically coherent representations.

Interestingly, this progression becomes more pronounced as model scale increases, with the 70B model showing the clearest separation between layer-specific functions. This indicates that larger models develop more specialized neural circuitry for processing complex cognitive states like regret, mirroring observations from our neuron intervention experiments.

## J.3 Thinking on Decoupling Patterning

**Oscillatory Decoupling Patterns analysis** According to complex systems theory Varley (2023), it think that Nervous systems involve multiple coupling and decoupling processes to achieve advanced function. Therefore, we observe that the oscillatory decoupling pattern in the regret mechanism is reasonable, as it is necessary for the decomposition and integration of information. Back to our regret research, attenton's mid- and high-level chaotic Ju et al. (2024) outputs provide direct evidence for the oscillatory decoupling pattern in Regret.

## J.4 Heuristic analogies with the brain

### J.4.1 Compositional Architecture and Brain Parallels: A Heuristic Analogy

**Analogy to Distributed Processing** The observed mutual information patterns between RegretD and DualD neurons in larger models (0.024-0.071 for 13B/70B vs. 0.015 for 7B) suggest a functional architecture that may be conceptually compared—as a purely heuristic analogy—to distributed processing in cognitive systems. While the implementation mechanisms differ fundamentally, this conceptual parallel offers an intuitive framework for understanding how regret emerges in LLMs. Our results indicate that LLMs process regret through interactions between functionally specialized neurons (RegretD) and multipurpose units (DualD), rather than isolated components. The consistently low mutual information between Non-RegretD and DualD neurons (0.007-0.047) across all model scales further supports this functional differentiation, with the 70B model demonstrating the clearest separation. This organizational principle of specialized components working in concert, rather

than in isolation, provides a useful conceptual framework for understanding emergent capabilities in large language models Rissman & Wagner (2012).

**Analogy to Combinatorial Neural Coding** Our findings can be conceptually related to principles of combinatorial neural coding Kim et al. (2025), where complex capabilities emerge from specific combinations of neural elements rather than isolated units. The progressive increase in RegretD-DualD mutual information ($0.015 \rightarrow 0.024 \rightarrow 0.071$) across model scales suggests that as models grow larger, they develop more integrated functional relationships between specialized neuron groups. This aligns with Kim et al.'s observation of combinatorial neural codes for long-term motor memory, although in a fundamentally different system context. This challenges simplistic interpretations of neural networks and highlights the importance of analyzing interaction patterns between neuron groups to understand complex capabilities like regret. The performance degradation observed when removing RegretD or DualD neurons (up to 4% drop in 7B models) provides empirical evidence for this combinatorial mechanism Kim et al. (2025). We emphasize that these analogies serve primarily as conceptual frameworks to guide our understanding of LLM architecture, rather than suggesting direct equivalence to biological systems.

### J.4.2 NON-MONOTONIC DYNAMICS: HEURISTIC ANALOGY TO BRAIN COGNITIVE DEVELOPMENT

Our findings on the non-monotonic scaling of regret processing in LLMs present *heuristic parallels* to principles observed in biological neural development. While direct mechanistic comparisons remain speculative, these analogies may offer conceptual bridges for understanding emergent phenomena in complex systems. We cautiously highlight two points of conceptual alignment:

**Critical integration as functional abstraction.** The surge in regret processing capabilities (MI $\geq$ 0.071 for RegretD-DualD in 70B models) suggests that complex functions emerge through *thresholds of compositional integration*. This loosely parallels findings where cognitive milestones (e.g., working memory maturation) require strengthened interactions between brain networks like the default mode network (DMN) and frontoparietal network (FPN) Chen et al. (2023a). However, we emphasize this as a *functional analogy*—while both systems exhibit integration-dependent emergence, the biological mechanisms (synaptic plasticity) differ fundamentally from artificial parameter optimization.

**Non-monotonicity as transitional states.** The performance dip in 13B models (Table. 3) heuristically mirrors non-linear trajectories in neurodevelopment. For instance, Qin et al. (2014) observed that hippocampal engagement in arithmetic learning first increases then decreases as cortical networks mature. Similarly, the 13B model's intermediate MI (0.024 vs. 70B's 0.071) may reflect an integration *transition phase*. These parallels invite exploration of *emergent modular synergy* across systems, though without implying equivalence in implementation.

However, while these heuristic parallels to cognitive development offer conceptual inspiration, we acknowledge limitations in our experimental approach to fully characterizing the non-monotonic scaling phenomena observed in this study. Unlike comprehensive developmental studies that can track changes across numerous stages, our analysis examined only three model scales (7B, 13B, 70B). Consequently, our findings represent preliminary observations rather than comprehensive scaling analysis. The interpretations we offer should be viewed as promising hypotheses for future investigation rather than definitive conclusions.

**Complementary Perspective on Scaling Laws** We emphasize that the non-monotonic scaling hypothesis represents a promising direction for future work that could potentially complement established scaling laws. Traditional scaling laws primarily focus on parameter count as the driving factor of performance, but our observations suggest architectural integration factors—specifically the mutual information between functional neuron groups—may play a crucial role not fully captured by parameter count alone. This perspective could help explain why certain capabilities emerge suddenly at specific model scales despite gradual parameter increases.

**Core Contributions and Next Steps** This limitation does not undermine our primary contributions—the regret analysis pipeline and compositional architecture findings—which are supported by our intervention experiments showing consistent effects across all tested model scales. Future work may extend our methodology to investigate scaling properties with finer granularity, potentially incorporating models trained with identical objectives but at more densely sampled parameter scales to

firmly establish the precise nature of these non-monotonic relationships. Additional research could also apply our analytical framework to other meta-cognitive capabilities beyond regret, potentially revealing whether similar compositional architectures underlie diverse cognitive functions in large language models.

### J.5 REASONABILITY ANALYSIS OF DATASET CONSTRUCTION

Our methodological framework for studying regret in LLMs rests on a solid foundation that effectively captures genuine internal mechanisms rather than artifacts. The strength of our approach derives from three interconnected elements:

**Human-Parallel Process Design** The design of our dataset parallels natural human error correction processes. While the *backfire effect* demonstrates that direct refutation may paradoxically reinforce erroneous beliefs in humans Nyhan & Reifler (2010), our methodology strategically induces regret through phased evidence exposure rather than confrontational correction. Specifically, humans typically express regret when contradictory evidence is presented with contextual scaffolding (e.g., reflection prompts)—a process distinct from adversarial belief challenges. By implementing our three-phase framework (fake evidence $\rightarrow$ hint cuing $\rightarrow$ real evidence presentation), we create an ecologically valid protocol that circumvents belief entrenchment while eliciting authentic meta-cognitive responses. Drawing inspiration from these human cognitive processes, we formalize regret in the context of LLMs through the following definition:

**Definition 1** (Regret in LLMs). Given a question $q$, information sets $\{I_i\}_{i=1}^n$, and responses $\{a_i\}_{i=1}^n$ where each $a_i$ is produced after receiving information set $I_i$, regret at step $i$ occurs when:

$$R_i(q, \{I_j\}_{j=1}^i, \{a_j\}_{j=1}^{i-1}) = \begin{cases} 1, & \text{if } a_i \text{ acknowledges regret for } a_1 \\ 0, & \text{otherwise} \end{cases}$$

Definition 1 formalizes our operational concept of regret in LLMs, providing a mathematical framework for systematic analysis. This definition captures the essential sequential nature of regret expression through information sets $\{I_i\}_{i=1}^n$ that directly correspond to our methodological stages: $I_1$ represents the fake evidence, $I_2$ introduces hint cuing, and $I_3$ provides real evidence. The model generates responses $\{a_i\}_{i=1}^n$ sequentially based on these cumulative information states, with regret ($R_i = 1$) manifesting when response $a_i$ explicitly acknowledges the error in $a_1$. This formalization enables precise identification and quantification of regret expressions as information states evolve throughout the experimental procedure.

**Autoregressive Integration and Signal Localization Strategy** Our approach leverages the fundamental autoregressive architecture of LLMs Touvron et al. (2023) to extract meaningful regret representations through explicit token anchoring. This methodological choice addresses three critical challenges in studying meta-cognitive states:

- **Contextual Integration:** Hidden states at "regret" tokens encapsulate the model's integrated processing of the complete interaction history—initial misinformation generation, hint-based reflection, and evidence-based correction—rather than isolated lexical encodings. For any token sequence where $x_i$ represents the $i$-th input token, the hidden state at position $t$ in layer $L$ is computed as $h_L^{(t)} = f_\theta(\{x_1, x_2, \ldots, x_t\})$, where $f_\theta$ denotes the transformer computation up to layer $L$. Thus, the regret token's hidden state contains compressed representations of the entire error-correction sequence, enabling analysis of the model's comprehensive internal representation of metacognitive error recognition.

- **Signal Anchoring Necessity:** Explicit token identification serves as a principled localization strategy in the absence of established benchmarks for LLM metacognition. This approach parallels successful interpretability studies that rely on specific token positions for systematic analysis (e.g., last subject tokens in factual recall Meng et al. (2022a), entity name tokens in spatial-temporal probing Gurnee & Tegmark (2023)). Recent layer-wise probing studies further demonstrate the effectiveness of token-specific analysis for understanding knowledge encoding Ju et al. (2024). Without such anchors, regret signals would be distributed across arbitrary token positions, making systematic neuron-level analysis intractable.

- **Concept Group Switch:** Our intervention experiments reveal that regret neurons function as conceptual group controllers. When we deactivate neurons identified through regret anchoring, LLM outputs show suppression not only of `regret` tokens but also related metacognitive expressions including `sorry`, `apologize`, `correction`, and `mistake`. This empirical observation indicates that regret-anchored neurons operate as conceptual switches controlling broader semantic networks rather than isolated lexical items. The systematic co-suppression of related terms suggests that our `regret` anchor successfully identifies neurons governing comprehensive metacognitive expression patterns, validating the broader applicability of our anchoring strategy beyond individual token analysis.

- **Methodological Validation:** We validate our approach through three convergent lines of evidence: (a) *Causal intervention*: Targeted neuron deactivation produces substantial performance degradation (up to 50.7%) compared to minimal effects from random neuron interventions, demonstrating genuine signal capture rather than spurious correlations; (b) *Cross-scale consistency*: The compositional architecture pattern (RegretD-DualD interactions) replicates across model scales, with mutual information showing non-monotonic scaling (7B:0.015$\rightarrow$13B:0.024$\rightarrow$70B:0.071) that reveals complex representational dynamics inconsistent with simple prompt artifacts; (c) *Differential activation*: RDS scores show clear distributional separation between regret and non-regret contexts, confirming signal specificity.

This explicit anchoring strategy, while necessarily limiting scope to overt regret expressions, establishes the foundational framework for analyzing LLM metacognitive mechanisms. Future extensions can leverage our S-CDI and GIC metrics to explore implicit regret through contextual inference, building upon the robust baseline established here.

## K  DATASET GENERATION PROMPT

---

**Generate Fake Evidence**

```
Generate fake evidence that supports the opposite of the following
ground truth: {ground_truth}. The evidence should be related to the
question: {question}. Ensure the evidence is false and misleading,
but still sounds plausible. Provide only the fake evidence, without
any additional explanations or introductions.
```

---

**Generate Initial Answer**

```
You previously answered the following question: {question}
Your initial answer was: {initial answer}
After considering the following hint: {weak hint}
Your revised answer was: {second answer}
Now, consider this information: {real evidence}
Which previous answer do you regret? What's the final answer?
Provide a direct answer in 1-5 sentences, focusing only on answering
the specific question.
```

---

**Generate Weak Hint**

```
The question is: {question}
The ground truth is: {ground_truth}
The following is fake evidence: {fake evidence}
The following is true evidence: {real evidence}
Your task is to generate a weak hint that subtly encourages the model
to reflect on the fake evidence.
The hint should meet the following criteria:
1. It should not directly reveal the correct answer or the true
evidence.
```

---

```
2.  It should not explicitly contradict the fake evidence.
3.  It should provide an indirect or metaphorical clue that might
lead the model to question the fake evidence.
4.  It should be neutral and open-ended, encouraging broader
thinking.
Provide only the weak hint, without any additional explanations or
introductions.
```

**Generate Second Answer**

```
You previously answered the following question:  {question}
Your initial answer was:  {initial answer}
Now, consider this hint:  {weak hint}
Do you regret your previous answer?  Provide a direct answer in 1-5
sentences, focusing only on answering the specific question.
```

**Generate Third Answer**

```
You previously answered the following question:  {question}
Your initial answer was:  {initial answer}
After considering the following hint:  {weak hint}
Your revised answer was:  {second answer}
Now, consider this information:  {real evidence}
Which previous answer do you regret?  What's the final answer?
Provide a direct answer in 1-5 sentences, focusing only on answering
the specific question.
```

## L  TERM EXPLANATION

This section provides comprehensive definitions of technical terms and methodological concepts introduced in our research on regret mechanisms in large language models.

**Core Metrics and Methods**

**S-CDI**  Supervised Compression-Decoupling Index. A metric that identifies optimal transformer layers where regret representations are most effectively decoupled from contextual features by balancing compression efficiency and class separability.

**RDS**  Regret Dominance Score. A neuron-level metric that quantifies the degree to which individual neurons are activated by regret versus non-regret contexts, enabling functional categorization of neurons into RegretD, Non-RegretD, and DualD groups.

**GIC**  Group Impact Coefficient. A metric that quantifies the functional impact of neuron groups both individually and compositionally through probe classification accuracy changes after neuron deactivation, revealing inter-group collaborative dynamics.

**CDI**  Compression-Decoupling Index. An unsupervised component of S-CDI that measures representation quality through feature redundancy and orthogonality, where lower values indicate more effective compression.

## Neuron Functional Categories

**RegretD**
Regret-Dominant neurons. Functional neuron category identified through RDS analysis, characterized by higher activation in regret contexts. These neurons serve as specialized processing units for regret-related representations and play critical roles in compositional regret architecture.

**Non-RegretD**
Non-Regret-Dominant neurons. Neuron category with higher activation in non-regret contexts, serving complementary functions to RegretD neurons. Their combination with other groups reveals compositional processing patterns.

**DualD**
Dual-function neurons. Neurons exhibiting balanced activation across both regret and non-regret contexts. These neurons play critical roles in compositional regret processing through collaborative interactions with RegretD neurons, particularly in larger models.

## Architectural and Processing Concepts

**Oscillatory Decoupling Pattern**
A systematic alternating pattern of coupling and decoupling phases across transformer layers revealed through S-CDI analysis. The pattern reflects the model's progression through feature entanglement, preliminary separation, contextual integration, and refined separation.

**Compositional Architecture**
The emergent organizational principle where regret representation relies on collaborative interactions between distinct neuron groups (RegretD, DualD, Non-RegretD) rather than isolated individual neurons. Validated through intervention experiments.

**Anchor-guided Gradient Attribution**
A cross-layer analysis methodology using RDS-identified functional neuron groups as attribution sources to discover regret-related neurons throughout the network, enabling targeted interventions across layers.

**Feature Entanglement**
The phenomenon where target representations (regret signals) are mixed with contextual, linguistic, and emotional features in neural activations. S-CDI analysis addresses this by identifying layers where regret features are optimally separated.

## Experimental and Validation Methods

**Neuron Intervention**
Experimental technique involving controlled activation suppression of specific neurons during forward propagation. Used to validate causal relationships between identified neuron groups and regret expression in model outputs.

**Probe Classifier**
A lightweight neural network (typically 2-layer MLP) trained on hidden states to detect regret-specific activation patterns. Serves as a diagnostic tool for evaluating regret signal strength across different layers.

**Hidden State Analysis**
Systematic examination of internal neural representations at specific token positions across transformer layers. In regret analysis, these serve as windows into the model's metacognitive processing.

**Mutual Information Analysis**
Statistical technique used to quantify information sharing between neuron groups. Reveals functional relationships between RegretD, DualD, and Non-RegretD neurons, with higher values indicating stronger collaborative interactions.

> **Key Notation**
>
> | | |
> |---|---|
> | $Z$ | Feature matrix with samples and feature dimensions |
> | $\ell, \ell^*$ | Layer indices, with $\ell^*$ denoting the optimal S-CDI layer |
> | $\tau$ | Threshold parameter for neuron categorization in RDS analysis |
> | $\beta$ | Intervention strength parameter for neuron deactivation experiments |
> | $\mu, \sigma$ | Mean and standard deviation for RDS score distributions |

## M  SOCIETAL IMPACT

This research on regret mechanisms in LLMs offers positive impacts through enhancing model reliability, improving interpretability, and developing more effective error correction techniques. However, potential negative impacts include the possibility of manipulating neurons to force false regret expressions. We believe understanding these mechanisms ultimately supports developing more reliable AI systems, while acknowledging that careful implementation is necessary.

## N  LIMITATIONS

The non-monotonic scaling observed in this paper is merely an interesting phenomenon that still lacks more detailed investigation. Our analysis was conducted on the LLaMA-2 model family (*7B, 13B, 70B*), which represents a well-established transformer architecture that provides sufficient scale diversity to demonstrate our core findings. While different frameworks and alignment techniques may influence internal representations, our work establishes a comprehensive research paradigm that includes the S-CDI metric, RDS categorization, and compositional analysis framework. Future researchers can readily adapt and generalize this paradigm to other model architectures to systematically investigate regret coding across diverse LLM families.

## O  LLMS USAGE IN THE PAPER

LLMs were used only occasionally to help polish the writing (propose new words, grammar and spelling correction). All technical ideas, experimental designs, analyses, conclusions, writing were developed and carried out entirely by the authors. The authors have full responsibility for the final text.

