# OpenReview forum: "Compositional Architecture of Regret in Large Language Models"
_ICLR.cc/2026/Conference — ICLR 2026 Conference Withdrawn Submission_

### Official Review · Reviewer_Woey · 2025-10-30

**Soundness:** 2
**Presentation:** 1
**Contribution:** 2
**Rating:** 2
**Confidence:** 4

**Summary:**

In this paper, the authors examine "regret" behavior in large language models from a mechanistic interpretability perspective. The work centers on defining and applying several metrics to identify neurons associated with regret expression. The authors propose a Supervised Compression-Decoupling Index (S-CDI) to locate transformer layers where regret-related features show separation from other features. Within the identified layers, a Regret Dominance Score (RDS) is calculated for individual neurons based on their activation patterns when the keyword "regret" appears in model outputs. Additionally, the authors define a Group Impact Coefficient (GIC) to measure relationships between different neuron groups. The experimental procedure follows this sequence: S-CDI identifies candidate layers, RDS categorizes neurons into three groups (RegretD, Non-RegretD, DualD), and GIC quantifies group interactions. The authors then conduct intervention experiments by deactivating specific neuron groups and measuring effects on model behavior.

**Strengths:**

- The paper focuses on an innovative problem and examines model behaviors that have not been previously explored in the literature.

- The paper provides substantial implementation details for each experimental step and process, facilitating reproducibility.

- The hierarchical strategy—progressing from layer-level to neuron-level analysis guided by quantitative metrics—is both intuitive and well-justified.

**Weaknesses:**

- The writing and structure of the paper could be improved. While it is commendable that the paper documents substantial details in the appendix, key steps and rationale critical for understanding the mechanistic interpretability approach—particularly the probing methodology—should be included in the main text, as how the method is applied is essential to interpret the results.  Moreover, the main text should primarily focus on providing complete high-level intuition rather than presenting dense technical details and cross-referencing essential information relegated to the appendix.

- The methodology and post-intervention evaluation rely heavily on explicit keyword matching. Specifically, regret is identified by probing hidden states at "regret" token positions, and the intervention effectiveness is assessed primarily through keyword presence. This approach risks capturing neurons that encode the semantics of the word "regret" rather than neurons that genuinely represent the metacognitive state of regret itself.
  - This keyword-centric design also raises concerns about the RDS metric's validity. Since RDS normalizes activations at "regret" tokens against activations at tokens from other words, the resulting scores may be disproportionately influenced by the activation magnitudes of these comparison tokens rather than capturing true functional specialization for regret processing.
  - As the neurons are often found to be highly polysementic, it is essential to show that the intervention effect is tightly manifest in "regret" behavior and not highly correlated to other behaviors.

- In the "Neuron Intervention: Single v.s. Compositional" experiment, the performance difference could also be explained by the number of neurons intervened. More experiments should be conducted to show that the performance difference is not (or weakly) related to the number of neurons involved.

- The method is subject to the choice of a different threshold (τ) as the values are different for each model scale.

- The analysis is restricted to the LLaMA-2 family, which undermines the finding's generalizability.

**Questions:**

Please see the above section.

---

### Official Review · Reviewer_jZ9Y · 2025-11-01

**Soundness:** 2
**Presentation:** 2
**Contribution:** 2
**Rating:** 2
**Confidence:** 4

**Summary:**

This paper explores the neural and representational basis of regret in large language models (LLMs). The authors consider a specific case of definition of regret where regret refers to an LLM’s explicit expression of self-correction i.e. when an LLM recognizes that its prior output was incorrect or misleading. The authors frame this as a form of artificial metacognition, where the model evaluates and revises its own outputs. To understand how regret is encoded, the authors look at which transformer layers most clearly encode regret signals, and how those signals are structured. They introduce three new metrics for understanding regret signals and build a novel dataset by prompting models with fake evidence -> hints -> true evidence, eliciting gradual regret responses. Using these tools, they find that regret is encoded via compositional cooperation among neuron groups, not isolated units, and that this encoding exhibits an oscillatory decoupling pattern across layers.

**Strengths:**

-The framing of regret is novel. Specifically, the authors treat regret as an emergent metacognitive feature, not a linguistic token.
- The work bridges interpretability, affective computing, and metacognition in LLMs.
- The metrics introduced are technically sound. RDS + GIC in particular form a hierarchical probing framework that moves beyond binary neuron relevance, addressing the combinatorial nature of cognitive features.
- The authors compare their approach across scales, metrics, and baselines and provide quantitative probe accuracies (~99%) and controlled interventions to substantiate findings.

**Weaknesses:**

- Regret is operationalized via lexical cues only (e.g., “I regret…”, “I apologize…”). This is a significant problem because it conflates linguistic form with cognitive metacognition. That is, the method identifies regret expressions, not internal evaluation per se.

- The dataset used relies heavily on GPT-4 synthetic scenarios and staged evidence conflicts. As a result, it may not capture naturally emergent regret; instead it may reflect learned politeness or self-correction behavior.

- While interventions suppress regret language, they don’t show that the same neurons generate the behavior. Suppression could arise from disrupting downstream linguistic coherence.

- Validation is limited. The study measures internal representations but not functional generalization: e.g., do the same neurons trigger regret across domains (math, dialogue, reasoning)?

- The method is only tested on LLaMA-2 models. As a result, there is no evidence that the discovered architecture transfers to GPT, PaLM, or instruction-tuned variants.

**Questions:**

1. How do you distinguish linguistic apology from true metacognitive regret in your dataset?
2. Could the same architecture apply to other self-referential states (e.g., doubt, confidence)?
3. Is the oscillatory decoupling pattern specific to transformer structure, or do you expect it in RNNs or Mixture-of-Experts models?
4. How sensitive are RDS and GIC to the choice of τ (threshold) and probe classifier?
5. Did you evaluate alternative decoupling metrics (e.g., mutual information directly)?
6. Does suppression of regret neurons affect model helpfulness or correction accuracy?
7. Could this approach be used to increase metacognitive behaviors (rather than suppress them)?

---

### Official Review · Reviewer_reuV · 2025-11-02

**Soundness:** 2
**Presentation:** 2
**Contribution:** 2
**Rating:** 4
**Confidence:** 3

**Summary:**

During our conversations with Large Language Models (LLMs), if we provide some factual information that contradicts the LLM’s previous outputs, then it might “regret” and change its mind. This paper analyzes the mechanism of the “regret” behavior. Due to an absence of specialized datasets capturing regret expressions, this paper first proposes a framework for constructing regret datasets. To solve the problem of a lack of metrics for identifying optimal layers and regret-related neurons, this paper proposes the Supervised Compression-Decoupling Index (S-CDI) to identify the optimal layer for regret representation, proposes the Regret Dominance Score (RDS) to identify regret-related neurons, and the Group Impact Coefficient (GIC) to measure the effect of regret-related neuron groups. Experiment results show an oscillatory pattern across layers, and show that neuron groups work together to induce the regret behavior.

**Strengths:**

(1) The motivation is interesting and worth studying. It is meaningful to understand how LLMs generate the regret behavior.

(2) The designed metrics are quite systematic and could help identify regret-related layers and neurons in that layer. Figure 3 shows the importance of designing each component.

(3) Experiment results show some interesting patterns.

**Weaknesses:**

(1) In the related work section, the second paragraph introduces related works for neuron probing. Lines 146-148 say, “While these existing approaches have advanced our understanding of how LLMs encode various linguistic features, there has been no quantitative analysis on which layers are the most important”. Actually, I think there were some quantitative analyses from existing probing research, such as [1, 2]. Paper [1] chose the layer where the probing vector is most effective. Paper [2] chose the layer where the two groups have the largest separation.

[1] Andy Arditi, Oscar Obeso, Aaquib Syed, Daniel Paleka, Nina Panickssery, Wes Gurnee, and Neel Nanda. Refusal in language models is mediated by a single direction, 2024.

[2] Lennart Burger, Fred A. Hamprecht, and Boaz Nadler. Truth is universal: Robust detection of lies in llms, 2024.

(2) The design of S-CDI seems a little ad hoc. I think it needs more justification. I am wondering why the compression efficiency is important. If we want to extract the layer that contributes the most to inducing the regretful behavior, then maybe compression efficiency is not important here. Besides, I didn’t fully understand why a lower S-CDI is better. A lower S-CDI indicates lower I_c(Z) and higher I_e(Z). It indicates that instances within the same class are less similar, and there is less separation between different classes, which means that the regret instances are less separated from the non-regret instances. I think in the optimal layer, the regret instances should be highly separable from non-regret instances. I am a little confusing here, and I am not sure whether my understanding is correct. Please let me know if my understanding is wrong.

(3) Lines 338-339 say, “Acc(Z − S) represents the classification accuracy after deactivating neurons in set S by setting their activation values to −1.” Maybe this deactivation method is not a standard approach. I think a more standard way is to set it to 0. If we set the activation values to -1, it might be out of the original representation distribution, which might lead to the model’s unexpected behaviors.

(4) Figure 4 (a) shows the probing accuracy and S-CDI. Except for the first layer, it seems that accuracy and S-CDI do not have a strong correlation. For layers after the first layer, when S-CDI changes, the accuracy almost stays the same. So I wonder whether S-CDI is a good metric to indicate the regret-related layer.

(5) Lines 395-396 say, “We will focus on the layer with the lowest S-CDI values (Last layer).” The last layer is directly connected to the output layer, so it is natural that the last layer contains the most related information to regret. It is more meaningful to study other layers.

(6) In the neuron intervention experiments, this paper finds that single-group intervention is not very effective, but combining RegretD with Non-RegretD or DualD is much more effective. Since Non-RegretD and DualD are marked as neurons less related to regret, these results make me wonder whether RDS is a good metric for identifying regret-related neurons.

(7) A minor issue: The presentation needs some improvement. There are many details in the appendix, but the main content lacks enough information to understand the high-level ideas of some experiments. For a paper with many designs and many experiments, it is natural to put details into the appendix, but the main content should at least include brief high-level ideas of the experiments.

(8) A minor clarity issue: Line 309 or 310 says, “To identify functionally distinct neuron subsets within Z”. But I didn’t find the definition of Z.

**Questions:**

Q1. In the definition of GIC (Equation 9), why is the denominator different for n=1 and n>=2?

Q2. Why does a lower S-CDI indicate a better decoupling effect?

---

### Official Review · Reviewer_Wdv4 · 2025-11-02

**Soundness:** 2
**Presentation:** 1
**Contribution:** 2
**Rating:** 2
**Confidence:** 3

**Summary:**

In this work, the authors investigate how large language models internally express regret-related signals. They introduce a new dataset designed to elicit and capture expressions of regret in model-generated text, with a particular emphasis on misinformation. The main objective is to use this dataset to identify which layers in the transformer correspond to regret signals and to analyze how these signals are represented within network activations. To pinpoint the relevant layer, the authors proposed a new metric called the supervised compression-decoupling index (S-CDI), which identifies the regret representation that is most distinct from other features. They then used a variant of the modality dominance score to categorize neurons into three clusters: regret, non-regret, and dual, and found clusters of neurons that collectively contribute to the emergence of regret representations based on group impact coefficients. They finally developed an intervention framework to verify their findings mechanistically based on gradient-based attribution.

**Strengths:**

The question the authors take up is definitely interesting and relevant.

They motivate the problem and their methods reasonably well.

**Weaknesses:**

Figures are way too small and fonts are not legible, formatting in general is not up to the standard of a typical ICLR publication.
I had to zoom 500% to see some of the figure labels and metrics.

The details of the pipeline is repeated multiple times from the intro, the figure description, and the method description. You can mention it once in more detail and keep the other explanations high level. This would also save safe for figures.

The schematic figure is a bit confusing, especially the last box of LLM responses with intervention. I was not able to understand the example in the box, could the authors please clarify this.

The details of gradient attributions and the final evaluation procedure is missing in the main text, which can brought back from the appendix.  This must be clarified.

There are too many results in the main paper, which makes it less focussed and hard to follow. I would try to focus on the main point and move supporting elements to the appendix. For example, the ablation figure can be placed in the appendix and mentioned briefly in the main text.

**Questions:**

Does it hold for other model families?

If you need to always intervene on combination of regret and non-regret/dual id to affect performance, what does that say about the regret neuron identification method?

Can the authors include one fully concrete working example in the main text? Something like figure 1 in this recent blogpost from Anthropic would be great: https://www.anthropic.com/research/introspection

---

### Note · Authors · 2025-11-14

I have read and agree with the venue's withdrawal policy on behalf of myself and my co-authors.